# Low Noise Short Wavelength Infrared Avalanche Photodetector Using SB-Based Strained Layer Superlattice

**Arash Dehzangi** [†] [iD]**, Jiakai Li** [†] **and Manijeh Razeghi** *

Center for Quantum Devices, Department of Electrical Engineering and Computer Science,
Northwestern University, Evanston, IL 60208, USA; arash.dehzangi@northwestern.edu (A.D.);
JiakaiLi2022@u.northwestern.edu (J.L.)
* Correspondence: razeghi@northwestern.edu
† These authors contributed equally.

**Abstract:** We demonstrate low noise short wavelength infrared (SWIR) Sb-based type II superlattice (T2SL) avalanche photodiodes (APDs). The SWIR GaSb/(AlAsSb/GaSb) APD structure was designed based on impact ionization engineering and grown by molecular beam epitaxy on a GaSb substrate. At room temperature, the device exhibits a 50% cut-off wavelength of 1.74 µm. The device was revealed to have an electron-dominated avalanching mechanism with a gain value of 48 at room temperature. The electron and hole impact ionization coefficients were calculated and compared to give a better prospect of the performance of the device. Low excess noise, as characterized by a carrier ionization ratio of ~0.07, has been achieved.

**Keywords:** short wavelength infrared; superlattice; avalanche photodiode; carrier ionization ratio





## 1. Introduction

Avalanche photodiodes (APDs) internally amplify charge carriers with an avalanche process while operating under a high reverse bias that can cause impact ionization compared to conventional p-i-n photodiodes. APDs can deliver a high sensitivity that is involved with a gain mechanism via avalanche multiplication, with several applications in the military and fiber-optic communication, imaging and commercial sectors [1–4].

For short wavelength infrared (SWIR) APDs, several material systems are implemented, including silicon, AlGaAs/InGaAsSb, InP/InGaAs, and HgCdTe (MCT) [5–7]. However, the spectral band between 1.5–2.6 µm of the SWIR range can be served further compared to InP/InGaAs or MCT. Lattice-matched InGaAs/InP can deliver high-performance APD devices operating in the 0.9-to-1.7 µm wavelength range. InGaAs detectors are capable of reaching longer cut-off wavelengths by increasing the indium content; however, the crystal defects introduced by the epitaxial process used to extend the indium content degrade the performance as the cut-off wavelength gets longer. MCT, on the other hand, is the most mature material system for infrared technology, but it suffers from drawbacks due to bulk and surface instability and due to higher costs, particularly for fabrication [8].

Due to the nature of impact ionization, the avalanche process is a random process, which is associated to a factor named excess noise F(M). According to the local-field avalanche theory, both the F(M) and the gain-bandwidth product of an APD can be impacted by the $k$ factor, which is the ratio of the hole ($\beta$) and electron ($\alpha$) ionization coefficients of the APD. As demonstrated by McIntyre [9], a large difference in the ionization rates for electrons and holes (low $k$ factor) is essential for a low noise avalanche photodiode. The F(M), which is given by

$$F(M) = kM + (1-k)\left(2 - \frac{1}{M}\right)$$

(1)

rises when increasing the gain (M), but the rate of increasing noise can be slowed down by reducing the $k$ value. A lower $k$ value can improve the performance of the APD devices [10]. Therefore, a low $k$ factor is crucial for a high-speed and low-noise operation of the APD device. The $k$ value can be minimized under single-carrier-initiated single-carrier multiplication (SCISCM) conditions (this means that an APD must be operated so that only one carrier species ionizes) [11,12]. This is difficult when for some materials the impact ionization coefficients are similar ($\beta/\alpha = k \cong 1$); it is therefore of great interest to explore the possibility of "artificially" decreasing $k$ in these materials by using APDs with bandstructure-engineered avalanche regions.

There is a great opportunity for introducing a new material capable of low dark current, high quantum efficiency, and single carrier multiplication for use in the strategic SWIR range. Antimony (Sb)-based III/V materials (bulk and superlattice) are capable of meeting the bandgap requirements for making APDs in the SWIR spectral range. In order to achieve great characteristics for Sb-based APDs with high gain and low noise, the bandgap and its electron and hole ionization coefficients have to be designed carefully. To minimize the excess noise factor, a pure or dominant electron or hole-initiated multiplication, along with an optimized hetero-junction design, can be applied via impact ionization engineering. One of the possible alternatives of impact ionization engineering for SWIR APDs is to use the multiquantum well (MQW) structure as the avalanche region. In the MQW-based avalanche region, the impact ionization happens easily between the heterointerfaces between the barrier and well layers due to sharp bandgap discontinuities [13].

Sb-based strained layer superlattice (SLS) material is [14] a developing material system with flexible band gap engineering and capabilities to cover the entire range of infrared light using different combination and compositions of Sb-based heterostructures, such as InAs/GaSb/AlSb or InAs/InAsSb with Type II staggered gap (type II) band alignment [15,16]. Recently new gain-based structures including APDs based on SLS Sb-base material have also been reported for the SWIR region [17,18].

The flexibility of T2SLs band structure engineering has a significant advantage for designing multiquantum well (MQW)-based APDs [19,20]. In this MQW structure, the band discontinuities between the well and barrier can be engineered to have a large conduction band discontinuity ($\Delta E_c$) and a small valence band discontinuity ($\Delta E_v$). In the MQW structure, the electron ionization rate can be enhanced since the electrons receive kinetic energy $\Delta E_c$ at hetero interfaces. Holes, on the other hand, can flow unhindered across the MQW because $\Delta E_v$ almost vanishes.

## 2. Materials and Methods

In this letter, we demonstrate a SWIR APD structure based on an MQW structure consisting of 20 loops of bulk GaSb well layer and AlAsSb/GaSb T2SL structure barrier layer sandwiched between two highly doped contact layers. The schematic of the design and structure of the SWIR APD device is shown in Figure 1a. The device structure was grown on 2-inch Te-doped n-type ($10^{17}$ cm$^{-3}$) GaSb (100) substrate using an Intevac Modular Gen II molecular beam epitaxy (MBE) system. As a first step, 100-nm-thick GaSb buffer layer was grown. Then, a 500-nm-thick n-contact ($10^{18}$ cm$^{-3}$) with InAs/GaSb/AlSb/GaSb SLS was grown. The superlattice design for the n-contact layer is similar to the design in our previous work [21]. Then, the absorption layer of 20 loops of undoped MQW consisting of AlAsSb/GaSb barrier layer and GaSb well layer was grown. The AlAs$_{0.10}$Sb$_{0.90}$ layer in the AlAsSb/GaSb superlattices is well lattice-matched to the GaSb substrate with antimony atoms in common with the substrate, which in turn can bring a great range of flexibility into the superlattice growth and design. At last, a 100-nm top p-contact ($10^{18}$ cm$^{-3}$) GaSb layer was grown. During growth, silicon and beryllium were used for the n-type and p-type dopant, respectively.

The empirical tight-binding method (ETBM) with *sps** formalism, with nearest neighbor interactions, under a two-center approximation, which was modified from previous work [22], was used to calculate the band discontinuities in the absorption region between

the AlAsSb/GaSb superlattice barrier and the GaSb well. The ETBM material parameter sets from the previous work were used [22] to calculate the band discontinuities in the absorption region between the AlAsSb/GaSb superlattice barrier and the GaSb well, and the design of the AlAsSb/GaSb superlattice was similar to what was previously reported [23]. The $\Delta E_c$ and $\Delta E_v$ between the barrier and well in the MQW structure were calculated to be ~0.50 eV and ~0.15 eV, respectively.

The energy-band diagram of the GaSb/(AlAsSb/GaSb) superlattice structure is shown in Figure 1b,c, with the unbiased and under bias voltages schematically illustrated. In this MQW structure, consider a hot electron accelerating in an AlAsSb/GaSb barrier layer under the bias voltage applied to the structure. When it enters in the GaSb well, it abruptly gains energy equal to the conduction band offset edge ($\Delta E_c$). The main effect is that the electron goes under a stronger electric field (increased by $\Delta E_c$). In contrast, the hole ionization rate $\beta$ is not substantially increased by the superlattice because the valence-band discontinuity is much smaller, which leads to a reduction in the *k* value.

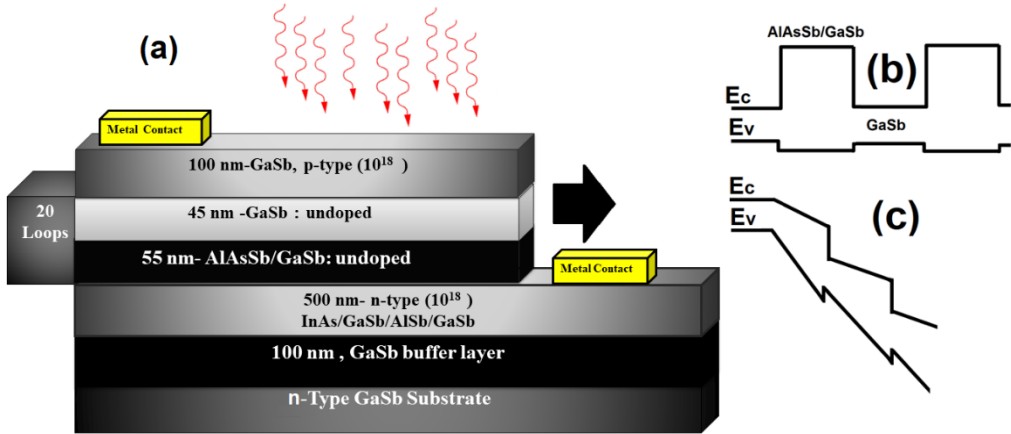

**Figure 1.** (**a**) Schematic of the SWIR APD structure under top illumination. Energy-band diagram of the MQW structure consisting of the AlAsSb/GaSb barrier layer and GaSb well layer: (**b**) unbiased (**c**) under bias voltages. The band edge discontinuities are $\Delta E_c \geq 0.5$ eV, $\Delta E_v < 0.15$ eV.

After the MBE growth, the material quality of the SWIR APD sample was assessed using atomic force microscopy (AFM) and high-resolution X-ray diffraction (HR-XRD). As shown in Figure 2a, the sample surface exhibits clear atomic steps with a small RMS roughness of 0.146 nm over a 10 μm × 10 μm area. Figure 2b shows the XRD scan curve of the APD sample, with the GaSb substrate and AlAsSb/GaSb superlattice. The mismatch between the AlAsSb/GaSb superlattice and the GaSb substrate is ~2500 ppm, while the AlAsSb layer exhibits a negative mismatch of ~−1000 ppm.

After the growth and characterizations, the material was then fabricated into circular photodetectors with mesa diameters ranging from 40 to 400 μm. Mesa-isolated etchings were performed by combined inductively coupled plasma (ICP) dry etching and citric-acid based wet etching technique. Both the top and bottom contacts were formed using electron beam-deposited Ti/Au. The devices were passivated for protection and insulation purposes by 1200 nm SiO$_2$ using plasma-enhanced chemical vapor deposition (PECVD). The detail about the fabrication can be found elsewhere [24]. Photodiodes were mounted onto a 68-pin leadless chip carrier (LCC) with indium for electrical and optical characterization.

The SWIR APD devices were optically characterized using a temperature- and pressure-controlled Janis STVP-100 two chamber liquid helium cryostat station with a 300 K background. The optical response of the SWIR APD was done under front-side illumination at room temperature. No antireflection coating was applied to the devices. The photodetector spectral response was measured using a Bruker IFS 66v/S Fourier transform infrared spectrometer (FTIR), and the absolute responsivity of the device was calculated using a band-pass filter in front of a blackbody source calibrated at 1000 °C.

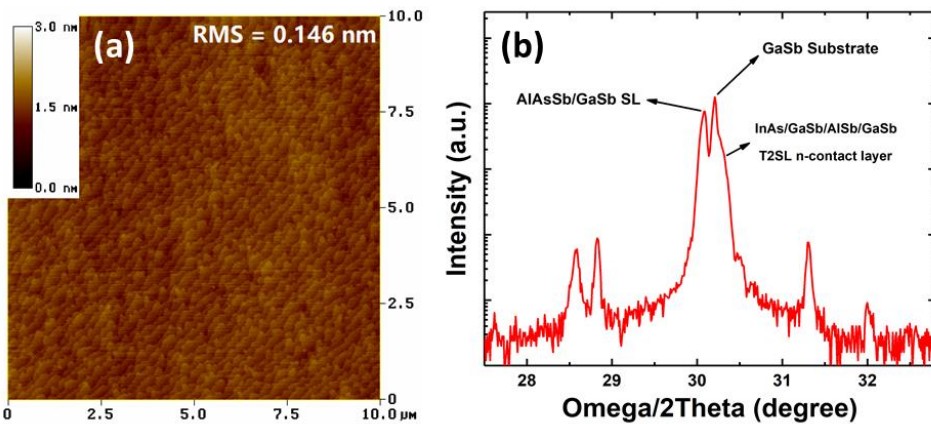

**Figure 2.** (**a**) The atomic force microscopy image of the grown MWIR APD sample over a 10 μm × 10 μm area. (**b**) HR-XRD scan curve of the grown APD sample, where peaks of different regions are marked.

## 3. Results

In order to verify the cut-off wavelength and draw a simple baseline for the optical performance of the device, the spectral response of the devices is shown in Figure 3. At room temperature, the device exhibits a 50% cut-off wavelength of 1.74 μm with the responsivity reaching a peak value of 0.38 A/W at 1.65 μm under a −5.0 V applied bias, respectively. Figure 3 also shows the photoluminescence of the grown wafer at room temperature with the peak of the spectrum being centered at λ = 1674 nm.

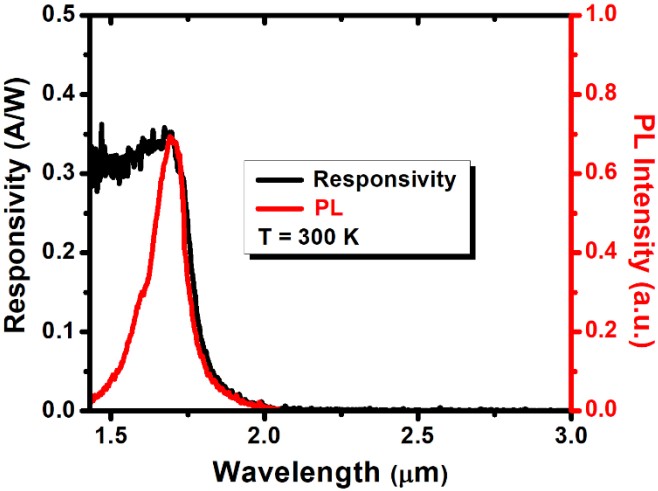

**Figure 3.** Responsivity spectra measured under front illumination at 300 K under −5.0 V bias voltage. (Red) The photoluminescence of the wafer at 300 K.

Current-voltage (I-V) measurements were carried out using an Agilent 4156c semiconductor parameter analyzer. The temperature of the devices was varied to study the change in gain characteristics at different temperatures. A 633 nm, an He-Ne laser with an incident power of 5.0 mW was used to measure the photocurrent and the gain of the APDs. Figure 4 shows the I-V characteristics of a 40-μm-sized SWIR APD with a gain of around 48 at a reverse bias voltage of −50 V at room temperature. The exponential nature of the gain indicates the occurrence of the avalanche mechanism, as seen in similar SWIR APDs. At 300 K, the device shows a unity optical gain dark current of $3.66 \times 10^{-6}$ A at a −19 V applied bias. The diodes show a punch-through effect at the voltage near ∼−19 V. This effect is considered to be related to the voltage required to achieve a full depletion

of the absorption layer; however, more studies in the future are needed to confirm this hypothesis for the present work.

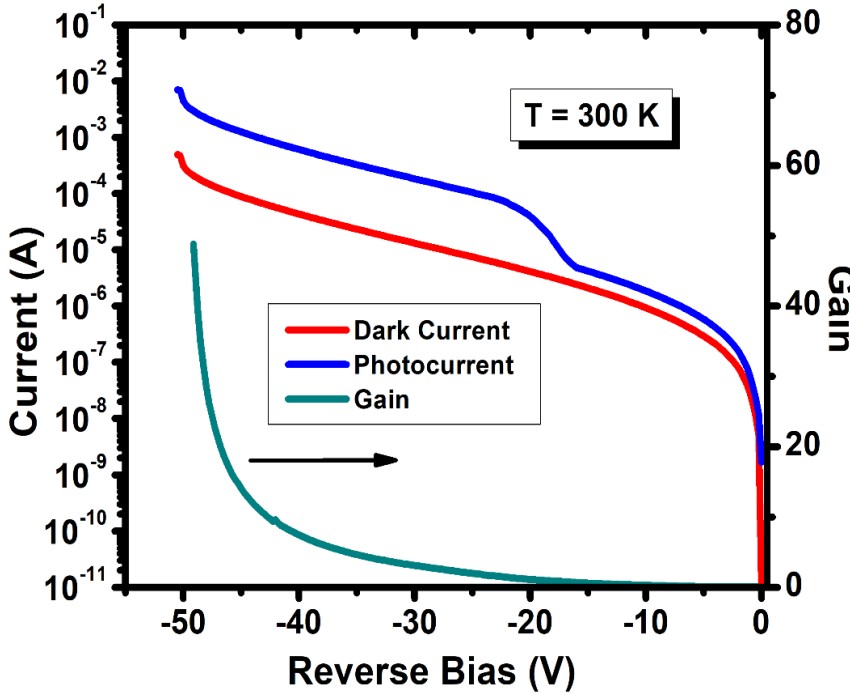

**Figure 4.** Breakdown characteristics and gain of the SWIR APD device at 300 K. The dark current and photocurrent are shown on the left axis; the gain is shown on the right axis.

We illuminated the device from either the top p+ contact or top n+ contact to control the dominant carrier injection into the multiplication region [25–27]. For the top contact illumination of the n+ contact, a separate device with a flipped structure was grown and processed under the same condition. In general, the electron and hole impact ionization coefficients $\alpha$ and $\beta$ can be derived from the experimental value of the electron-initiated avalanche gain and hole-initiated avalanche gain ($M_e$ and $M_h$) by solving the avalanche rate equations [28]. The extracted electron and hole impact ionization rate for SWIR APD is shown in Figure 5. In order to give a better prospect of the present work by comparing it to other reported cases, Figure 5 also presents a comparison of the obtained hole and electron ionization coefficients for SWIR APD vs. the inverse of the electric field at T = 300 K with the reported coefficients for AlGaAsSb and AlGaAs [29,30]. It is worth noting that unlike the other material systems the impact ionization value for the electrons stays high and unchanged at a lower electric field ($8 \times 10^{-6}$ cm/V corresponding to 125 kV/cm) range, and the cause of this trend is subject to further study for a future research direction.

The large difference in $\alpha$ and $\beta$ (see Figure 5) under these different injection regimes is direct evidence that the effective $\alpha$ is greater than $\beta$. It also implies that the avalanche multiplication process is dominated by the impact ionization of electrons. The carrier ionization ratio for the SWIR APD was calculated to be 0.07 = $k$ at room temperature (300 K). This small $k$ value is largely due to an enhanced electron impact ionization, which also agrees well with the theoretical predictions and experimental results of this effect in a superlattice with characteristics similar to ours [13,19].

The gain of the device was measured at different temperatures, as illustrated in Figure 6. At 200 K, the gain of the SWIR APD device reached around 206 at a −50 V bias voltage and stayed constant up to 220 K before showing a continuous slow decrease at higher temperatures upon warming up to room temperature. The decrease of the gain value at a higher temperature could be associated to a higher probability of carrier−carrier scattering and higher loss of the kinetic energy of carriers via various scattering mechanisms, and more studies are needed in this subject.

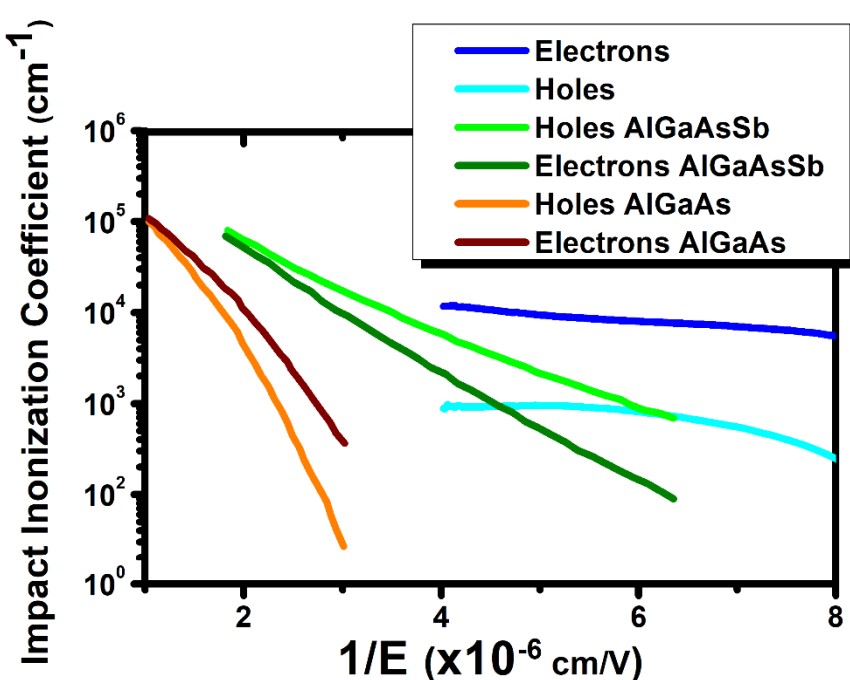

**Figure 5.** A comparison of the obtained hole and electron ionization coefficients for SWIR APD vs. the inverse of the electric field at T= 300 K with the reported coefficients for AlGaAsSb and AlGaAs [29,30].

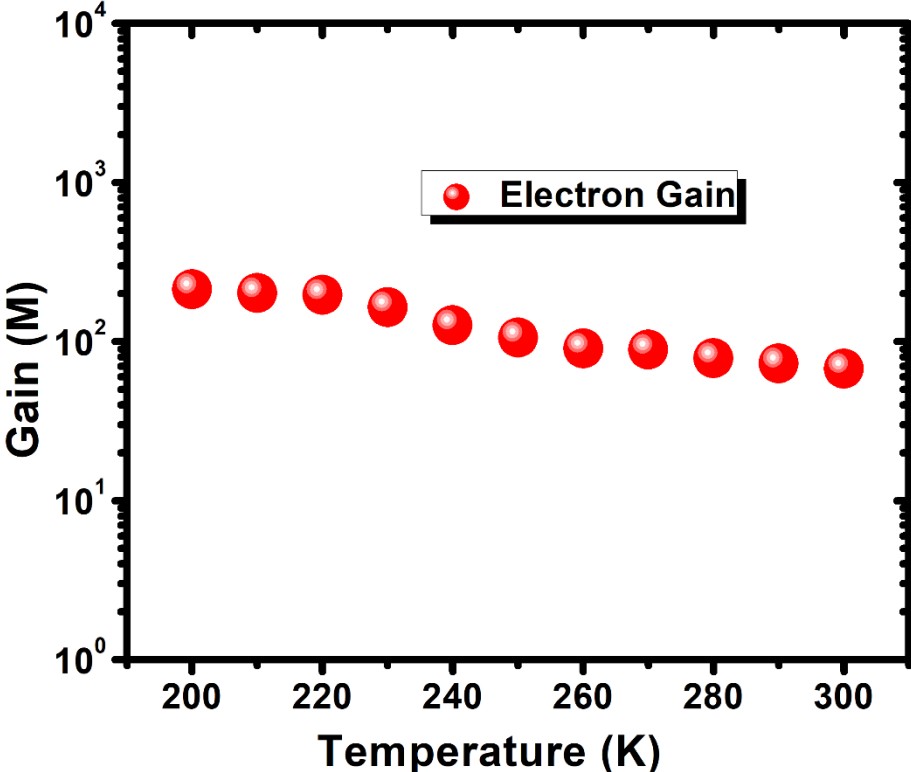

**Figure 6.** Temperature-dependent gain characteristics of the SWIR APD from 220 K to 300 K at −50 V bias voltage.

Figure 7 shows the excess noise factors measured for the SWIR APD (using an SR770 FFT spectrum analyzer) and the comparison with the excess noise values predicted by the local field model in Equation (1) [9]. For lower values of the multiplication gain in the

gain range of 1 to 20, the measured SWIR APD excess noise corresponds to an estimated *k*-value of 0.07, which was expected; however, for higher gain values of 20 to 50, there is a small increasing trend as the multiplication gain increases in correspondence to the *k*-value between 0.07 and 0.1. This could be due to a limitation of the measurement system at a high biased voltage and yet it needs to be further studied for future research directions.

This *k* value is close to the best reported values for APD devices based on similar materials, such as AlInAsSb (*k* = 0.015) [17], AlAsSb (*k* = 0.1) [31], and AlGaAs (*k* = 0.1), and it is better than the *k* values for commercial Silicon bases APDs (*k* values of around 0.02 and 0.06) and InP-based APDs (*k* values between 0.4 and 0.5) and InAlAs (*k* values in the range of 0.2 and 0.3) [31]. Through the further development of the device architecture and implementing band engineering in the superlattice system, a low noise-based SWIR device based on SLS material is achievable for a low noise application with a higher gain bandwidth product.

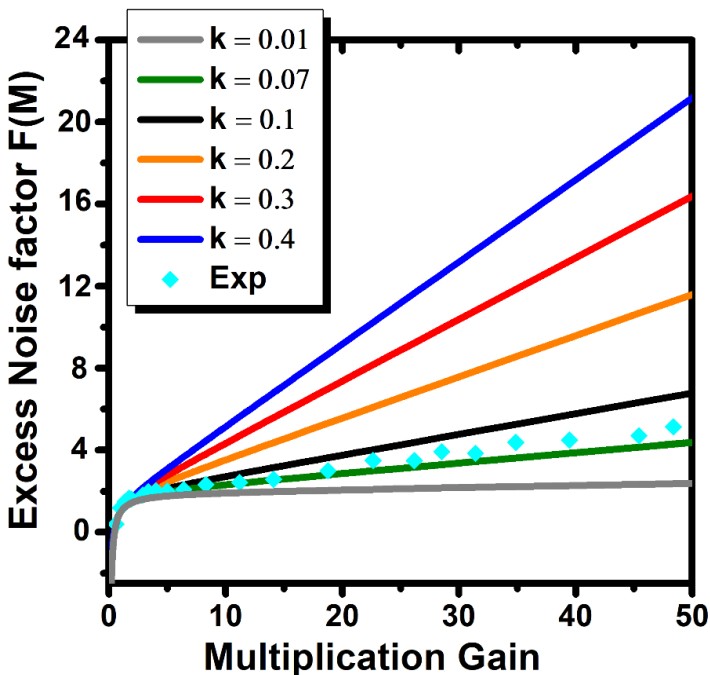

**Figure 7.** Measured excess noise factor at T = 300 K versus gain for the SWIR APD. The solid lines are plots of the excess noise factor using the local field model for *k* values from 0.01 to 0.4.

### 4. Conclusions

In summary, using impact ionization engineering, an Sb-based SLS material structure was implemented to demonstrate a low-noise SWIR GaSb/(AlAsSb/GaSb) superlattice APD device. A multiplication gain of 48 was achieved at room temperature. The structure was designed based on impact ionization engineering by implementing the MQW structure. The device exhibits a 50% cut-off wavelength of 1.74 μm at room temperature. The electron and hole impact ionization coefficients for the SWIR APD device were calculated and compared with each other to give a better prospect of the performance. This led to extracting a carrier ionization ratio with a value of 0.07 for the SWIR APD. The SWIR APDs revealed promising gain/noise characteristics for low noise applications.

**Author Contributions:** A.D. worked on the design, did the fabrication and measurements of the devices and wrote the whole paper; J.L. helped for analyzing the data; M.R. supervised the project. All authors have read and agreed to the published version of the manuscript.

**Funding:** This work was partially supported by the Army Research lab under (W911NF1810402), and the authors would like to acknowledge the support and interest of Tania Paskova from US Army Future Command.

**Institutional Review Board Statement:** Not applicable.

**Informed Consent Statement:** Not applicable.

**Data Availability Statement:** The data that support the findings of this study are available from the corresponding author upon reasonable request.

**Acknowledgments:** The authors would like to acknowledge the interest and encouragement of Jay Lewis and Whitney Mason from DARPA, Michael Gerhold and Tania Paskova from the U.S. Army Futures Command, Kurt Eyink from the Air Force Research Laboratory, Murzy Jhabvala from the NASA Goddard Space Flight Center, Meimei Tidrow from the U.S. Army Night Vision Laboratory, Manzur Tariq CIV USN NUWC DIV NEWPORT RI (USA) from the Navy, Carolynn Moor from the Army, and Myron Pauli from the Navy.

**Conflicts of Interest:** The authors declare no conflict of interest.

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
