# Peer review of "Low Noise Short Wavelength Infrared Avalanche Photodetector Using SB-Based Strained Layer Superlattice"

_photonics, doi:10.3390/photonics8050148_

Round 1

Reviewer 1 Report

"Due to the nature of impact ionization the avalanche process is a random process, hence it renders a high frequency noise at the device output"
 * It's not clear what this means. Excess noise is measured across the frequency spectrum, as it was in this manuscript using an SR770. Hence, it's white noise. What is "high frequency" about it? 

"According to the local-field avalanche theory, both the F(M) and the gain-bandwidth product of an APD are determined by the k factor"
 * The gain-bandwidth product isn't necessarily determined by k. The impact ionization ratio can set an upper limit, but the gain-bandwidth product is honestly outside of this manuscript's scope. It's usually RC limited at low gain. See Nature Photonics Vol 3, pg 59 for an example of the different limiting mechanisms. The manuscripts walks back this claim a few sentences later: "lower k value also can improve the gain bandwidth product." Or more clearly, in one instance the gain bandwidth product "is determined" by k and the other it "can improve" gain bandwidth product.

"The exponential nature of the gain indicates single carrier dominated avalanche mechanism..." 
 * All APD gain is exponential with electric field. It does not necessarily indicate single carrier gain.

Provide more details about the superlattice, including exact layer thicknesses and compositions. 
 * Is this even a T2SL? AlSb/GaSb is not a T2SL, as the band alignment is Type 1. There isn't enough information to tell from the AlAsSb.
 * Note that there are numerous papers covering AlGaAsSb at this point in time. Your addition to the field is through the specific superlattice. More information is warranted.

"The structural properties of the superlattices were evaluated after growth using high resolution X-ray diffraction (HRXRD) and atomic force microscopy (AFM)."
 * Please provide the results of those experiments, including lattice mismatch and RMS roughness across an area. 

The paper claims that "circular photodetectors with mesa diameters ranging from 40 to 400 μm" were fabricated, but there's no data on this range of areas. The manuscript contradicts itself a page later, when the characterized device was a 40 um x 40 um device, i.e. square. Are there sidewall leakage paths? Are the devices squares or circles? While the geometric distinction appears pointless, there are numerous references that examine the effects of high electric fields on corners. And assuming that these devices are meant for an array, that has potential implications. 

Why is the spectral responsivity plotted at -5 VDC? The APD has not reached punch through yet (i.e. the "unity optical gain" point) and so it's difficult to determine whether this accurately represents its spectrum. This measurement should be done at higher reverse bias to capture the APD behavior.
 * I strongly recommend using the term "punch through" in your manuscript. It's a more standard term.
 * C-V measurements would be helpful to back up the "unity optical gain" claim.

The impact ionization coefficient plot in Figure 4 relies upon careful measurement of both the gain and the electric field. While the electron gain measurement is supported by the data provided, the electric field is not. A C-V measurement is almost certainly required to support this calculation, especially since the superlattice is unlikely to have fully depleted. That's evidenced by both the well/barrier design and the 10x improvement in photocurrent around -18 VDC.
 * It would be helpful to plot the impact ionization coefficients of AlGaAsSb from APL Vol 104, Pg 162103 and/or APL Vol 112 pg 021103 on the same chart. I understand that the structures are different, but the novelty of the manuscript is to gauge the effects of superlattices on impact ionization. 
 * Since you fabricated a flipped structure, please comment on the IV curves and gain curves. Were they identical? Were they different? What was the unity optical gain dark current? What was the maximum gain? 
 * What are the maximum and minimum bias values used in Figure 4? 
 * What are the A, E_c, and m values for the impact ionization coefficients? 

The references are not acceptable in current form. Please remove many of the self citations. This manuscript does not provide much, if any, information about superlattice development. Instead, it covers APDs. The authors appear to be purposefully citing their work in a marginally related field at the expense of APD researchers who have contributed far more to this research area. Notable author omissions include John David, Erik Duerr, Majeed Hayat, Andrew Huntington, Andrew Marshall, and Chee Hing Tan. 

Author Response

Dear Editor,

All the reviewers raised interesting points relating to the results presented in the paper.  We have carefully reviewed each comment and made the necessary modifications to the paper.  Please find enclosed our report to the comments of the referee.

Reviewer 2 Report

1) Fig 1(b) and Fig 1 (c): the energy-band diagrams for unbiased and biased structures should be placed one below the other for better understanding.

2) at the line 118: "circular photodetectors mesa diameter 40 to 400 micromrters" and at line 147: "40x40 µm2 SWIR APD", my question: what is actual shape of the device?

3) ref [3] - incomplete data of the reference

Author Response

(The authors gave the same response as above.)

Reviewer 3 Report

In the paper titled Low noise short–wavelength infrared avalanche photodetector using Sb-based strained layer superlattice authors presented interesting results of Sb containing SL avalanche photodiode infrared detectors. Presented results are very interesting especially for researchers working with  infrared radiation.

However, before the publication in Photonics authors have to address some issues listed below:

  • please include references to the sentence, for example line 27: fiber-optic communication, imaging and commercial sector.[1-3]  should be changed to fiber-optic communication, imaging and commercial sector [1-3].
  • 2. Materials and Methods - in the text authors mentioned that buffer thickness is 200 nm while in Fig. 1 buffer is 100 nm thick
  • How authors verified doping concentration in contact layers?
  • Please use thinner lines in Fig. 2 and smaller point in Fig. 5
  • Please refer to results and parameter values of other groups working on APD detectors when discussing their parameters, for example (lines 194-195): The values of the  excess noise factor are promising compared to the values reported for III-V based SWIR APDs. [21, 31, 32] - what values?

Author Response

(The authors gave the same response as above.)

Round 2

Reviewer 1 Report

Thank you for addressing most of my comments. There are still two that have not been addressed.

1. Please provide the exact dimensions per layer in nm or monolayers for both the AlAsSb/GaSb barriers and the InAs/GaSb/AlSb/GaSb contacts, as previously requested. There is no scientific reason to omit this information, which is critical for reproducibility.

2. A more thorough discussion of the impact ionization coefficients is necessary without CV data. It's very difficult to understand how the impact ionization coefficient for electrons could be 1E4 per inv. cm at field strengths of 125 kV/cm, when it's so much higher than anything previously reported. While it's acceptable to not include CV measurements, Figure 5 raises questions that must be either stated or addressed. A sentence or two should be sufficient.

Author Response

Reviewer #1:

Comments and Suggestions for Authors

  • A more thorough discussion of the impact ionization coefficients is necessary without CV data. It's very difficult to understand how the impact ionization coefficient for electrons could be 1E4 per inv. cm at field strengths of 125 kV/cm, when it's so much higher than anything previously reported. While it's acceptable to not include CV measurements, Figure 5 raises questions that must be either stated or addressed. A sentence or two should be sufficient.

Answer:

The trend was addressed in the manuscript. As mentioned before, in depth study of the performance of device including CV performance is presently under further study at the center for quantum device. 

Reviewer 3 Report

Authors wrote: We have carefully reviewed each comment and made the necessary modifications to the paper.

It is not true. For example I asked authors to include references into the sentence. Reference is a part of the sentence so it have to be before the dot which ends sentence.

Author Response

Comments and Suggestions for Authors

  • Please include references to the sentence, for example line 27: fiber-optic communication, imaging and commercial sector. [1-3]  should be changed to fiber-optic communication, imaging and commercial sector [1-3].

Answer:

Thanks for reminding us about the references, in revised version we corrected the format accordingly.

Round 3

Reviewer 1 Report

The authors have sufficiently responded to all comments and suggestions. This manuscript is acceptable for publication.

Reviewer 3 Report

I am satisfied with the paper improvement.